# MODEL-AUGMENTED ACTOR-CRITIC: BACKPROPAGATING THROUGH PATHS

## ABSTRACT

Current model-based reinforcement learning approaches use the model simply as a learned black-box simulator to augment the data for policy optimization or value function learning. In this paper, we show how to make more effective use of the model by exploiting its differentiability. We construct a policy optimization algorithm that uses the pathwise derivative of the learned model and policy across future timesteps. Instabilities of learning across many timesteps are prevented by using a terminal value function, learning the policy in an actor-critic fashion. Furthermore, we present a derivation on the monotonic improvement of our objective in terms of the gradient error in the model and value function. We show that our approach (i) is consistently more sample efficient than existing state-of-the-art model-based algorithms, (ii) matches the asymptotic performance of model-free algorithms, and (iii) scales to long horizons, a regime where typically past model-based approaches have struggled.

## 1 INTRODUCTION

Model-based reinforcement learning (RL) offers the potential to be a general-purpose tool for learning complex policies while being sample efficient. When learning in real-world physical systems, data collection can be an arduous process. Contrary to model-free methods, model-based approaches are appealing due to their comparatively fast learning. By first learning the dynamics of the system in a supervised learning way, it can exploit off-policy data. Then, model-based methods use the model to derive controllers from it either parametric controllers (Luo et al., 2019; Buckman et al., 2018; Janner et al., 2019) or non-parametric controllers (Nagabandi et al., 2017; Chua et al., 2018).

Current model-based methods learn with an order of magnitude less data than their model-free counterparts while achieving the same asymptotic convergence. Tools like ensembles, probabilistic models, planning over shorter horizons, and meta-learning have been used to achieved such performance (Kurutach et al., 2018; Chua et al., 2018; Clavera et al., 2018). However, the model usage in all of these methods is the same: simple data augmentation. They use the learned model as a black-box simulator generating samples from it. In high-dimensional environments or environments that require longer planning, substantial sampling is needed to provide meaningful signal for the policy. *Can we further exploit our learned models?*

In this work, we propose to estimate the policy gradient by backpropagating its gradient through the model using the pathwise derivative estimator. Since the learned model is differentiable, one can link together the model, reward function, and policy to obtain an analytic expression for the gradient of the returns with respect to the policy. By computing the gradient in this manner, we obtain an expressive signal that allows rapid policy learning. We avoid the instabilities that often result from back-propagating through long horizons by using a terminal Q-function. This scheme fully exploits the learned model without harming the learning stability seen in previous approaches (Kurutach et al., 2018; Heess et al., 2015). The horizon at which we apply the terminal Q-function acts as a hyperparameter between model-free (when fully relying on the Q-function) and model-based (when using a longer horizon) of our algorithm.

The main contribution of this work is a model-based method that significantly reduces the sample complexity compared to state-of-the-art model-based algorithms (Janner et al., 2019; Buckman et al., 2018). For instance, we achieve a 10k return in the half-cheetah environment in just 50 trajectories. We theoretically justify our optimization objective and derive the monotonic improvement of our

learned policy in terms of the Q-function and the model error. Furtermore, we experimentally analyze the theoretical derivations. Finally, we pinpoint the importance of our objective by ablating all the components of our algorithm. The results are reported in four model-based benchmarking environments (Wang et al., 2019; Todorov et al., 2012). The low sample complexity and high performance of our method carry high promise towards learning directly on real robots.

## 2 RELATED WORK

**Model-Based Reinforcement Learning.** Learned dynamics models offer the possibility to reduce sample complexity while maintaining the asymptotic performance. For instance, the models can act as a learned simulator on which a model-free policy is trained on (Kurutach et al., 2018; Luo et al., 2019; Janner et al., 2019). The model can also be used to improve the target value estimates (Feinberg et al., 2018) or to provide additional context to a policy (Du & Narasimhan, 2019). Contrary to these methods, our approach uses the model in a different way: we exploit the fact that the learned simulator is differentiable and optimize the policy with the analytical gradient. Long term predictions suffer from a compounding error effect in the model, resulting in unrealistic predictions. In such cases, the policy tends to overfit to the deficiencies of the model, which translates to poor performance in the real environment; this problem is known as model-bias (Deisenroth & Rasmussen, 2011). The model-bias problem has motivated work that uses meta-learning (Clavera et al., 2018), interpolation between different horizon predictions (Buckman et al., 2018; Janner et al., 2019), and interpolating between model and real data (Kalweit & Boedecker, 2017). To prevent model-bias, we exploit the model for a short horizon and use a terminal value function to model the rest of the trajectory. Finally, since our approach returns a stochastic policy, dynamics model, and value function could use model-predictive control (MPC) for better performance at test time, similar to (Lowrey et al., 2018; Hong et al., 2019). MPC methods (Nagabandi et al., 2017) have shown to be very effective when the uncertainty of the dynamics is modelled (Chua et al., 2018; Wang & Ba, 2019).

**Differentable Planning.** Previous work has used backpropagate through learned models to obtain the optimal sequences of actions. For instance, Levine & Abbeel (2014) learn linear local models and obtain the optimal sequences of actions, which is then distilled into a neural network policy. The planning can be incorporated into the neural network architecture (Okada et al., 2017; Tamar et al., 2016; Srinivas et al., 2018; Karkus et al., 2019) or formulated as a differentiable function (Pereira et al., 2018; Amos et al., 2018). Planning sequences of actions, even when doing model-predictive control (MPC), does not scale well to high-dimensional, complex domains Janner et al. (2019). Our method, instead learns a neural network policy in an actor-critic fashion aided with a learned model. In our study, we evaluate the benefit of carrying out MPC on top of our learned policy at test time, Section 5.4. The results suggest that the policy captures the optimal sequence of action, and re-planning does not result in significant benefits.

**Policy Gradient Estimation.** The reinforcement learning objective involves computing the gradient of an expectation (Schulman et al., 2015a). By using Gaussian processes (Deisenroth & Rasmussen, 2011), it is possible to compute the expectation analytically. However, when learning expressive parametric non-linear dynamical models and policies, such closed form solutions do not exist. The gradient is then estimated using Monte-Carlo methods (Mohamed et al., 2019). In the context of model-based RL, previous approaches mostly made use of the score-function, or REINFORCE estimator (Peters & Schaal, 2006; Kurutach et al., 2018). However, this estimator has high variance and extensive sampling is needed, which hampers its applicability in high-dimensional environments. In this work, we make use of the pathwise derivative estimator (Mohamed et al., 2019). Similar to our approach, Heess et al. (2015) uses this estimator in the context of model-based RL. However, they just make use of real-world trajectories that introduces the need of a likelihood ratio term for the model predictions, which in turn increases the variance of the gradient estimate. Instead, we entirely rely on the predictions of the model, removing the need of likelihood ratio terms.

**Actor-Critic Methods.** Actor-critic methods alternate between policy evaluation, computing the value function for the policy; and policy improvement using such value function (Sutton & Barto, 1998; Barto et al., 1983). Actor-critic methods can be classified between on-policy and off-policy. On-policy methods tend to be more stable, but at the cost of sample efficiency (Sutton, 1991; Mnih et al., 2016). On the other hand, off-policy methods offer better sample complexity (Lillicrap et al., 2015). Recent work has significantly stabilized and improved the performance of off-policy methods using maximum-entropy objectives (Haarnoja et al., 2018a) and multiple value functions (Fujimoto

et al., 2018). Our method combines the benefit of both. By using the learned model we can have a learning that resembles an on-policy method while still being off-policy.

## 3 BACKGROUND

In this section, we present the reinforcement learning problem, two different lines of algorithms that tackle it, and a summary on Monte-Carlo gradient estimators.

### 3.1 REINFORCEMENT LEARNING

A discrete-time finite Markov decision process (MDP) $\mathcal{M}$ is defined by the tuple $(\mathcal{S}, \mathcal{A}, f, r, \gamma, p_0, T)$. Here, $\mathcal{S}$ is the set of states, $\mathcal{A}$ the action space, $s_{t+1} \sim f(s_t, a_t)$ the transition distribution, $r : \mathcal{S} \times \mathcal{A} \to \mathbb{R}$ is a reward function, $p_0 : \mathcal{S} \to \mathbb{R}_+$ represents the initial state distribution, $\gamma$ the discount factor, and $T$ is the horizon of the process. We define the return as the sum of rewards $r(s_t, a_t)$ along a trajectory $\tau := (s_0, a_0, ..., s_{T-1}, a_{T-1}, s_T)$. The goal of reinforcement learning is to find a policy $\pi_\theta : \mathcal{S} \times \mathcal{A} \to \mathbb{R}^+$ that maximizes the expected return, i.e., $\max_\theta J(\theta) = \max_\theta \mathbb{E}[\sum_t \gamma^t r(s_t, a_t)]$.

**Actor-Critic.** In actor-critic methods, we learn a function $\hat{Q}$ (critic) that approximates the expected return conditioned on a state $s$ and action $a$, $\mathbb{E}[\sum_t \gamma^t r(s_t, a_t)|s_0 = s, a_0 = a]$. Then, the learned Q-function is used to optimize a policy $\pi$ (actor). Usually, the Q-function is learned by iteratively minimizing the Bellman residual:

$$J_Q = \mathbb{E}[(\hat{Q}(s_t, a_t) - (r(s_t, a_t) + \gamma \hat{Q}(s_{t+1}, a_{t+1})))^2]$$

The above method is referred as one-step Q-learning, and while a naive implementation often results in unstable behaviour, recent methods have succeeded in stabilizing the Q-function training (Fujimoto et al., 2018). The actor then can be trained to maximize the learned $\hat{Q}$ function $J_\pi = \mathbb{E}\left[\hat{Q}(s, \pi(s))\right]$. The benefit of this form of actor-critic method is that it can be applied in an off-policy fashion, sampling random mini-batches of transitions from an experience replay buffer (Lin, 1992).

**Model-Based RL.** Model-based methods, contrary to model-free RL, learn the transition distribution from experience. Typically, this is carried out by learning a parametric function approximator $\hat{f}_\phi$, known as a dynamics model. We define the state predicted by the dynamics model as $\hat{s}_{t+1}$, i.e., $\hat{s}_{t+1} \sim \hat{f}_\phi(s_t, a_t)$. The models are trained via maximum likelihood: $\max_\phi J_f(\phi) = \max_\phi \mathbb{E}[\log p(\hat{s}_{t+1}|s_t, a_t)]$

### 3.2 MONTE-CARLO GRADIENT ESTIMATORS

In order to optimize the reinforcement learning objective, it is needed to take the gradient of an expectation. In general, it is not possible to compute the exact expectation so Monte-Carlo gradient estimators are used instead. These are mainly categorized into three classes: the pathwise, score function, and measure-valued gradient estimator (Mohamed et al., 2019). In this work, we use the pathwise gradient estimator, which is also known as the re-parameterization trick (Kingma & Welling, 2013). This estimator is derived from the law of the unconscious statistician (LOTUS) (Grimmett & Stirzaker, 2001)

$$\mathbb{E}_{p_\theta(x)}[f(x)] = \mathbb{E}_{p(\epsilon)}[f(g_\theta(\epsilon)]$$

Here, we have stated that we can compute the expectation of a random variable $x$ without knowing its distribution, if we know its corresponding sampling path and base distribution. A common case, and the one used in this manuscript, $\theta$ parameterizes a Gaussian distribution: $x \sim p_\theta = \mathcal{N}(\mu_\theta, \sigma_\theta^2)$, which is equivalent to $x = \mu_\theta + \epsilon \sigma_\theta$ for $\epsilon \sim \mathcal{N}(0, 1)$.

## 4 POLICY GRADIENT VIA MODEL-AUGMENTED PATHWISE DERIVATIVE

Exploiting the full capability of learned models has the potential to enable complex and high-dimensional real robotics tasks while maintaining low sample complexity. Our approach, model-augmented actor-critic (MAAC), exploits the learned model by computing the analytic gradient

of the returns with respect to the policy. In contrast to sample-based methods, which one can think of as providing directional derivatives in trajectory space, MAAC computes the full gradient, providing a strong learning signal for policy learning, which further decreases the sample complexity. In the following, we present our policy optimization scheme and describe the full algorithm.

### 4.1 MODEL-AUGMENTED ACTOR-CRITIC OBJECTIVE

Among model-free methods, actor-critic methods have shown superior performance in terms of sample efficiency and asymptotic performance (Haarnoja et al., 2018a). However, their sample efficiency remains worse than model-based approaches, and fully off-policy methods still show instabilities comparing to on-policy algorithms (Mnih et al., 2016). Here, we propose a modification of the Q-function parametrization by using the model predictions on the first time-steps after the action is taken. Specifically, we do policy optimization by maximizing the following objective:

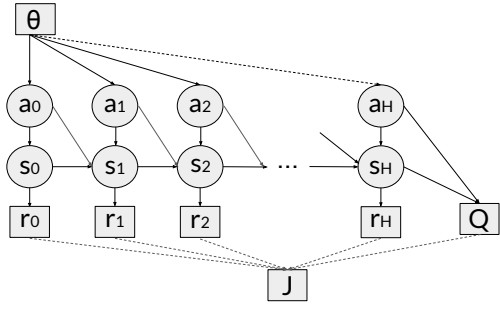

Figure 1: Stochastic computation graph of the proposed objective $J_\pi$. The stochastic nodes are represented by circles and the deterministic ones by squares.

$$J_\pi(\boldsymbol{\theta}) = \mathbb{E}\left[\sum_{t=0}^{H-1} \gamma^t r(s_t) + \gamma^H \hat{Q}(s_H, a_H)\right]$$

whereby, $s_{t+1} \sim \hat{f}(s_t, a_t)$ and $a_t \sim \pi_{\boldsymbol{\theta}}(s_t)$. Note that under the true dynamics and Q-function, this objective is the same as the RL objective. Contrary to previous reinforcement learning methods, we optimize this objective by back-propagation through time. Since the learned dynamics model and policy are parameterized as Gaussian distributions, we can make use of the pathwise derivative estimator to compute the gradient, resulting in an objective that captures uncertainty while presenting low variance. The computational graph of the proposed objective is shown in Figure 1.

While the proposed objective resembles n-step bootstrap (Sutton & Barto, 1998), our model usage fundamentally differs from previous approaches. First, we do not compromise between being off-policy and stability. Typically, n-step bootstrap is either on-policy, which harms the sample complexity, or its gradient estimation uses likelihood ratios, which presents large variance and results in unstable learning (Heess et al., 2015). Second, we obtain a strong learning signal by backpropagating the gradient of the policy across multiple steps using the pathwise derivative estimator, instead of the REINFORCE estimator (Mohamed et al., 2019; Peters & Schaal, 2006). And finally, we prevent the exploding and vanishing gradients effect inherent to back-propagation through time by the means of the terminal Q-function (Kurutach et al., 2018).

The horizon $H$ in our proposed objective allows us to trade off between the accuracy of our learned model and the accuracy of our learned Q-function. Hence, it controls the degree to which our algorithm is model-based or well model-free. If we were not to trust our model at all ($H = 0$), we would end up with a model-free update; for $H = \infty$, the objective results in a shooting objective. Note that we will perform policy optimization by taking derivatives of the objective, hence we require accuracy on the derivatives of the objective and not on its value. The following lemma provides a bound on the gradient error in terms of the error on the derivatives of the model, the Q-function, and the horizon $H$.

**Lemma 4.1** (Gradient Error). *Let $\hat{f}$ and $\hat{Q}$ be the learned approximation of the dynamics $f$ and Q-function $Q$, respectively. Assume that $Q$ and $\hat{Q}$ have $L_q/2$-Lipschitz continuous gradient and $f$ and $\hat{f}$ have $L_f/2$-Lipschitz continuous gradient. Let $\epsilon_f = \max_t \|\nabla \hat{f}(\hat{s}_t, \hat{a}_t) - \nabla f(s_t, a_t)\|_2$ be the error on the model derivatives and $\epsilon_Q = \|\nabla \hat{Q}(\hat{s}_H, \hat{a}_H) - \nabla Q(s_H, a_H)\|_2$ the error on the Q-function derivative. Then the error on the gradient between the learned objective and the true objective can be bounded by:*

$$\mathbb{E}\left[\|\nabla_{\boldsymbol{\theta}} J_\pi - \nabla_{\boldsymbol{\theta}} \hat{J}_\pi\|_2\right] \leq c_1(H)\epsilon_f + c_2(H)\epsilon_Q$$

*Proof.* See Appendix. □

The result in Lemma 4.1 stipulates the error of the policy gradient in terms of the maximum error in the model derivatives and the error in the Q derivatives. The functions $c_1$ and $c_2$ are functions of the horizon and depend on the Lipschitz constants of the model and the Q-function. Note that we are just interested in the relation between both sources of error, since the gradient magnitude will be scaled by the learning rate, or by the optimizer, when applying it to the weights.

## 4.2 MONOTONIC IMPROVEMENT

In the previous section, we presented our objective and the error it incurs in the policy gradient with respect to approximation error in the model and the Q function. However, the error on the gradient is not indicative of the effect of the desired metric: the average return. Here, we quantify the effect of the modeling error on the return. First, we will bound the KL-divergence between the policies resulting from taking the gradient with the true objective and the approximated one. Then we will bound the performance in terms of the KL.

**Lemma 4.2** (Total Variation Bound). *Under the assumptions of the Lemma 4.1, let $\boldsymbol{\theta} = \boldsymbol{\theta}_o + \alpha \nabla_{\boldsymbol{\theta}} J_\pi$ be the parameters resulting from taking a gradient step on the exact objective, and $\hat{\boldsymbol{\theta}} = \boldsymbol{\theta}_o + \alpha \nabla_{\boldsymbol{\theta}} \hat{J}_\pi$ the parameters resulting from taking a gradient step on approximated objective, where $\alpha \in \mathbb{R}^+$. Then the following bound on the total variation distance holds*

$$\max_s D_{TV}(\pi_\theta || \pi_{\hat{\theta}}) \leq \alpha c_3 (\epsilon_f c_1(H) + \epsilon_Q c_2(H))$$

*Proof.* See Appendix. □

The previous lemma results in a bound on the distance between the policies originated from taking a gradient step using the true dynamics and Q-function, and using its learned counterparts. Now, we can derive a similar result from Kakade & Langford (2002) to bound the difference in average returns.

**Theorem 4.1** (Monotonic Improvement). *Under the assumptions of the Lemma 4.1, be $\boldsymbol{\theta}'$ and $\hat{\boldsymbol{\theta}}$ as defined in Lemma 4.2, and assuming that the reward is bounded by $r_{\max}$. Then the average return of the $\pi_{\hat{\theta}}$ satisfies*

$$J_\pi(\hat{\boldsymbol{\theta}}) \geq J_\pi(\boldsymbol{\theta}) - \frac{2\alpha r_{\max}}{1 - \gamma} \alpha c_3 (\epsilon_f c_1(H) + \epsilon_Q c_2(H))$$

*Proof.* See Appendix. □

Hence, we can provide explicit lower bounds of improvement in terms of model error and function error. Theorem 4.1 extends previous work of monotonic improvement for model-free policies (Schulman et al., 2015b; Kakade & Langford, 2002), to the model-based and actor critic set up by taking the error on the learned functions into account. From this bound one could, in principle, derive the optimal horizon $H$ that minimizes the gradient error. However, in practice, approximation errors are hard to determine and we treat $H$ as an extra hyper-parameter. In section 5.2, we experimentally analyze the error on the gradient for different estimators and values of $H$.

## 4.3 ALGORITHM

Based on the previous sections, we develop a new algorithm that explicitly optimizes the model-augmented actor-critic (MAAC) objective. The overall algorithm is divided into three main steps: model learning, policy optimization, and Q-function learning.

**Model learning.** In order to prevent overfitting and overcome model-bias (Deisenroth & Rasmussen, 2011), we use a bootstrap ensemble of dynamics models $\{\hat{f}_{\phi_1}, ..., \hat{f}_{\phi_M}\}$. Each of the dynamics models parameterizes the mean and the covariance of a Gaussian distribution with diagonal covariance. The bootstrap ensemble captures the epistemic uncertainty, uncertainty due to the limited capacity or data, while the probabilistic models are able to capture the aleatoric uncertainty (Chua et al., 2018), inherent uncertainty of the environment. We denote by $f_\phi$ the transitions dynamics resulting from $\phi_U$, where $U \sim \mathcal{U}[M]$ is uniform random variable on $\{1, ..., M\}$. The dynamics models are trained via maximum likelihood with early stopping on a validation set.

---

**Algorithm 1** MAAC

---

1: Initialize the policy $\pi_{\boldsymbol{\theta}}$, model $\hat{f}_{\boldsymbol{\phi}}$, $\hat{Q}_{\boldsymbol{\psi}}$, $\mathcal{D}_{\text{env}} \leftarrow \emptyset$, and the model dataset $\mathcal{D}_{\text{model}} \leftarrow \emptyset$
2: **repeat**
3:    Sample trajectories from the real environment with policy $\pi_{\theta}$. Add them to $\mathcal{D}_{\text{env}}$.
4:    **for** $i = 1 \ldots G_1$ **do**
5:       $\boldsymbol{\phi} \leftarrow \boldsymbol{\phi} - \beta_f \nabla_{\boldsymbol{\phi}} J_f(\boldsymbol{\phi})$ using data from $\mathcal{D}_{\text{env}}$.
6:    **end for**
7:    Sample trajectories $\mathcal{T}$ from $\hat{f}_{\boldsymbol{\phi}}$. Add them to $\mathcal{D}_{\text{model}}$.
8:    $\mathcal{D} \leftarrow \mathcal{D}_{\text{model}} \cup \mathcal{D}_{\text{env}}$
9:    **for** $i = 1 \ldots G_2$ **do**
10:       Update $\boldsymbol{\theta} \leftarrow \boldsymbol{\theta} + \beta_{\pi} \nabla_{\boldsymbol{\theta}} J_{\pi}(\boldsymbol{\theta})$ using data from $\mathcal{D}$
11:       Update $\boldsymbol{\psi} \leftarrow \boldsymbol{\psi} - \beta_Q \nabla_{\boldsymbol{\psi}} J_Q(\boldsymbol{\psi})$ using data from $\mathcal{D}$
12:    **end for**
13: **until** the policy performs well in the real environment
14: **return** Optimal parameters $\boldsymbol{\theta}^*$

---

**Policy Optimization.** We extend the MAAC objective with an entropy bonus (Haarnoja et al., 2018b), and perform policy learning by maximizing

$$J_{\pi}(\boldsymbol{\theta}) = \mathbb{E}\left[\sum_{t=0}^{H-1} \gamma^t r(\hat{s}_t) + \gamma^H Q_{\boldsymbol{\psi}}(\hat{s}_H, a_H)\right] + \beta \mathcal{H}(\pi_{\boldsymbol{\theta}})$$

where $\hat{s}_{t+1} \sim f_{\boldsymbol{\phi}}(\hat{s}_t, a_t)$, $\hat{s}_0 \sim \mathcal{D}$, $a \sim \pi_{\boldsymbol{\theta}}$. We learn the policy by using the pathwise derivative of the model through $H$ steps and the Q-function by sampling multiple trajectories from the same $\hat{s}_0$. Hence, we learn a maximum entropy policy using pathwise derivative of the model through $H$ steps and the Q-function. We compute the expectation by sampling multiple actions and states from the policy and learned dynamics, respectively.

**Q-function Learning.** In practice, we train two Q-functions (Fujimoto et al., 2018) since it has been experimentally proven to yield better results. We train both Q functions by minimizing the Bellman error (Section 3.1):

$$J_Q(\boldsymbol{\psi}) = \mathbb{E}[(Q_{\boldsymbol{\psi}}(s_t, a_t) - (r(s_t, a_t) + \gamma Q_{\boldsymbol{\psi}}(s_{t+1}, a_{t+1})))^2]$$

Similar to (Janner et al., 2019), we minimize the Bellman residual on states previously visited and imagined states obtained from unrolling the learned model. Finally, the value targets are obtained in the same fashion the Stochastic Ensemble Value Expansion (Buckman et al., 2018), using $H$ as a horizon for the expansion. In doing so, we maximally make use of the model by not only using it for the policy gradient step, but also for training the Q-function.

Our method, MAAC, iterates between collecting samples from the environment, model training, policy optimization, and Q-function learning. A practical implementation of our method is described in Algorithm 1. First, we obtain trajectories from the real environment using the latest policy available. Those samples are appended to a replay buffer $\mathcal{D}_{\text{env}}$, on which the dynamics models are trained until convergence. The third step is to collect imaginary data from the models: we collect $k$-step transitions by unrolling the latest policy from a randomly sampled state on the replay buffer. The imaginary data constitutes the $\mathcal{D}_{\text{model}}$, which together with the replay buffer, is used to learn the Q-function and train the policy.

Our algorithm consolidates the insights built through the course of this paper, while at the same time making maximal use of recently developed actor-critic and model-based methods. All in all, it consistently outperforms previous model-based and actor-critic methods.

## 5   RESULTS

Our experimental evaluation aims to examine the following questions: 1) How does MAAC compares against state-of-the-art model-based and model-free methods? 2) Does the gradient error correlate

with the derived bound?, 3) Which are the key components of its performance?, and 4) Does it benefit from planning at test time?

In order to answer the posed questions, we evaluate our approach on model-based continuous control benchmark tasks in the MuJoCo simulator (Todorov et al., 2012; Wang et al., 2019).

## 5.1 COMPARISON AGAINST STATE-OF-THE-ART

We compare our method on sample complexity and asymptotic performance against state-of-the-art model-free (MF) and model-based (MB) baselines. Specifically, we compare against the model-free soft actor-critic (SAC) (Haarnoja et al., 2018a), which is an off-policy algorithm that has been proven to be sample efficient and performant; as well as two state-of-the-art model-based baselines: model-based policy-optimization (MBPO) (Janner et al., 2019) and stochastic ensemble value expansion (STEVE) (Buckman et al., 2018). The original STEVE algorithm builds on top of the model-free algorithm DDPG (Lillicrap et al., 2015), however this algorithm is outperformed by SAC. In order to remove confounding effects of the underlying model-free algorithm, we have implemented the STEVE algorithm on top of SAC. We also add SVG(1) (Heess et al., 2015) to comparison, which similar to our method uses the derivative of dynamic models to learn the policy.

The results, shown in Fig. 2, highlight the strength of MAAC in terms of performance and sample complexity. MAAC scales to higher dimensional tasks while maintaining its sample efficiency and asymptotic performance. In all the four environments, our method learns faster than previous MB and MF methods. We are able to learn near-optimal policies in the half-cheetah environment in just over 50 rollouts, while previous model-based methods need at least the double amount of data. Furthermore, in complex environments, such as ant, MAAC achieves near-optimal performance within 150 rollouts while other take orders of magnitudes more data.

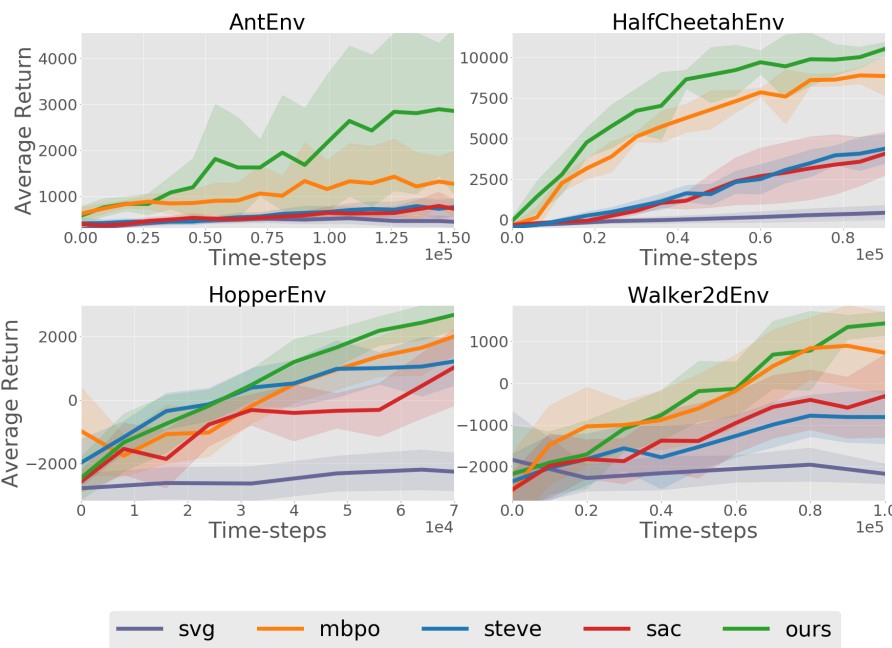

Figure 2: Comparison against state-of-the-art model-free and model-based baselines in four different MuJoCo environments. Our method, MAAC, attains better sample efficiency and asymptotic performance than previous approaches. The gap in performance between MAAC and previous work increases as the environments increase in complexity.

## 5.2 GRADIENT ERROR

Here, we investigate how the bounds obtained relate to the empirical performance. In particular, we study the effect of the horizon of the model predictions on the gradient error. In order to do so, we

construct a double integrator environment; since the transitions are linear and the cost is quadratic for a linear policy, we can obtain the analytic gradient of the expect return.

Figure 3 depicts the $L1$ error of the MAAC objective for different values of the horizon $H$ as well as what would be the error using the true dynamics. As expected, using the true dynamics yields to lower gradient error since the only source comes from the learned Q-function that is weighted down by $\gamma^H$. The results using learned dynamics correlate with our assumptions and the derived bounds: the error from the learned dynamics is lower than the one in the Q-funtion, but it scales poorly with the horizon. For short horizons the error decreases as we increase the horizon. However, large horizons is detrimental since it magnifies the error on the models.

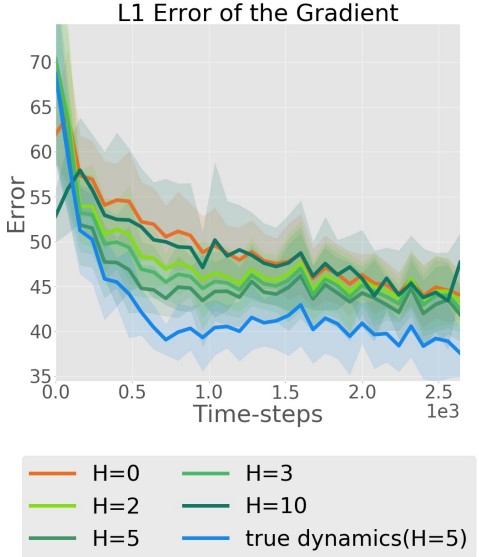

### 5.3 ABLATIONS

In order to investigate the importance of each of the components of our overall algorithm, we carry out an ablation test. Specifically, we test three different components: 1) not using the model to train the policy, i.e., set $H = 0$, 2) not using the STEVE targets for training the critic, and 3) using a single sample estimate of the path-wise derivative.

The ablation test is shown in Figure 4. The test underpins the importance of backpropagating through the model: setting $H$ to be 0 inflicts a severe drop in the algorithm performance. On the other hand, using the STEVE targets results in slightly more stable training, but it does not have a significant effect. Finally, while single sample estimates can be used in simple environments, they are not accurate enough in higher dimensional environments such as ant.

Figure 3: $L1$ error on the policy gradient when using the proposed objective for different values of the horizon $H$ as well as the error obtained when using the true dynamics. The results correlate with the assumption that the error in the learned dynamics is lower than the error in the Q-function, as well as they correlate with the derived bounds.

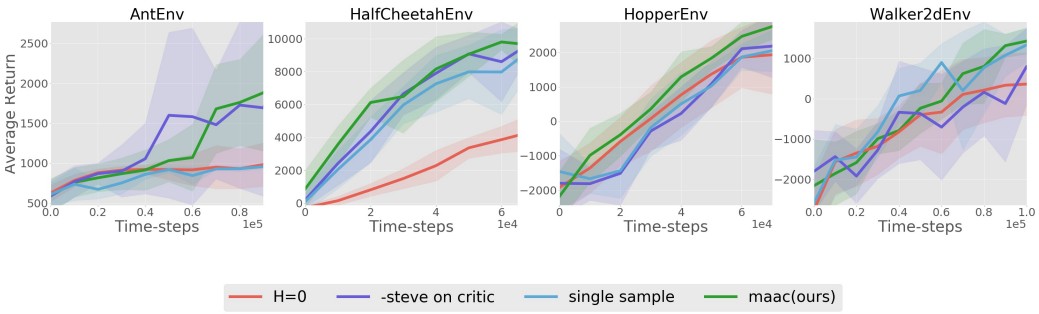

Figure 4: Ablation test of our method. We test the importance of several components of our method: not using the model to train the policy ($H = 0$), not using the STEVE targets for training the Q-function (-STEVE), and using a single sample estimate of the pathwise derivative. Using the model is the component that affects the most the performance, highlighting the importance of our derived estimator.

### 5.4 MODEL PREDICTIVE CONTROL

One of the key benefits of methods that combine model-based reinforcement learning and actor-critic methods is that the optimization procedure results in a stochastic policy, a dynamics model and a Q-function. Hence, we have all the components for, at test time, refine the action selection by the

means of model predictive control (MPC). Here, we investigate the improvement in performance of planning at test time. Specifically, we use the cross-entropy method with our stochastic policy as our initial distributions. The results, shown in Table 2, show benefits in online planning in complex domains; however, its improvement gains are more timid in easier domains, showing that the learned policy has already interiorized the optimal behaviour.

| | AntEnv | HalfCheetahEnv | HopperEnv | Walker2dEnv |
|---|---|---|---|---|
| MAAC+MPC | $3.97e3 \pm 1.48e3$ | $1.09e4 \pm 94.5$ | $2.8e3 \pm 11$ | $1.76e3 \pm 78$ |
| MAAC | $3.06e3 \pm 1.45e3$ | $1.07e4 \pm 253$ | $2.77e3 \pm 3.31$ | $1.61e3 \pm 404$ |

Table 1: Performance at test time with (maac+mpc) and without (maac) planning of the converged policy using the MAAC objective.

## 6 CONCLUSION

In this work, we present model-augmented actor-critic, MAAC, a reinforcement learning algorithm that makes use of a learned model by using the pathwise derivative across future timesteps. We prevent instabilities arisen from backpropagation through time by the means of a terminal value function. The objective is theoretically analyzed in terms of the model and value error, and we derive a policy improvement expression with respect to those terms. Our algorithm that builds on top of MAAC is able to achieve superior performance and sample efficiency than state-of-the-art model-based and model-free reinforcement learning algorithms. For future work, it would be enticing to deploy the presented algorithm on a real-robotic agent.

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

# A APPENDIX

Here we prove the lemmas and theorems stated in the manuscript.

## A.1 PROOF OF LEMMA 4.1

Let $J_\pi(\boldsymbol{\theta})$ and $\hat{J}_\pi(\hat{\boldsymbol{\theta}})$ be the expected return of the policy $\pi_{\boldsymbol{\theta}}$ under our objective and under the RL objective, respectively. Then, we can write the MSE of the gradient as

$$\mathbb{E}[\|\nabla_{\boldsymbol{\theta}} J_\pi(\boldsymbol{\theta}) - \nabla_{\boldsymbol{\theta}} \hat{J}_\pi(\boldsymbol{\theta})\|_2] = \mathbb{E}[\|\nabla_{\boldsymbol{\theta}}(M - \hat{M}) + |\nabla_{\boldsymbol{\theta}} \gamma^H (Q - \hat{Q})\|_2]$$
$$\leq \mathbb{E}[\|\nabla_{\boldsymbol{\theta}}(M - \hat{M})\|_2] + \mathbb{E}[\|\nabla_{\boldsymbol{\theta}} \gamma^H (Q - \hat{Q})\|_2]$$

whereby, $M = \sum_{t=0}^H \gamma^t r(s_t)$ and $\hat{M} = \sum_{t=0}^H \gamma^t r(\hat{s}_t)$.

We will denote as $\nabla$ the gradient w.r.t the inputs of network, $x_t = (s_t, a_t)$ and $\hat{x}_t = (\hat{s}_t, \hat{a}_t)$; where $\hat{a}_t \sim \pi_{\boldsymbol{\theta}}(\hat{s}_t)$. Notice that since $f\hat{f}$ and $\pi$ are Lipschitz and their gradient is Lipschitz as well, we have that $\nabla_{\boldsymbol{\theta}} \hat{x}_t \leq K^t$, where K depends on the Lipschitz constants of the model and the policy. Without loss of generality, we assume that K is larger than 1. Now, we can bound the error on the Q as

$$\|\nabla_{\boldsymbol{\theta}}(Q - \hat{Q})\|_2 = \|\nabla Q \nabla_{\boldsymbol{\theta}} x_H - \nabla \hat{Q} \nabla_{\boldsymbol{\theta}} \hat{x}_H\|_2$$
$$= \|(\nabla Q - \nabla \hat{Q}) \nabla_{\boldsymbol{\theta}} x_H - \nabla \hat{Q}(\nabla_{\boldsymbol{\theta}} \hat{x}_H - \nabla_{\boldsymbol{\theta}} x_H)\|_2$$
$$\leq \|\nabla Q - \nabla \hat{Q}\|_2 \|\nabla_{\boldsymbol{\theta}} x_H\|_2 + \|\nabla \hat{Q}\|_2 \|\nabla_{\boldsymbol{\theta}} \hat{x}_H - \nabla_{\boldsymbol{\theta}} x_H\|_2$$
$$\leq \epsilon_Q \|\nabla_{\boldsymbol{\theta}} x_H\|_2 + L_Q \|\nabla_{\boldsymbol{\theta}} \hat{x}_H - \nabla_{\boldsymbol{\theta}} x_H\|_2$$
$$\leq \epsilon_Q K^H + L_Q \|\nabla_{\boldsymbol{\theta}} \hat{x}_H - \nabla_{\boldsymbol{\theta}} x_H\|_2$$

Now, we will bound the term $\|\nabla_{\boldsymbol{\theta}} \hat{s}_{t+1} - \nabla_{\boldsymbol{\theta}} s_{t+1}\|_2$:

$$\|\nabla_{\boldsymbol{\theta}} \hat{s}_{t+1} - \nabla_{\boldsymbol{\theta}} s_{t+1}\|_2 = \|\nabla_s \hat{f} \nabla_{\boldsymbol{\theta}} \hat{s}_t + \nabla_a \hat{f} \nabla_{\boldsymbol{\theta}} \hat{a}_t - \nabla_s f \nabla_{\boldsymbol{\theta}} s_t - \nabla_a f \nabla_{\boldsymbol{\theta}} a_t\|_2$$
$$\leq \|\nabla_s \hat{f} \nabla_{\boldsymbol{\theta}} \hat{s}_t - \nabla_s f \nabla_{\boldsymbol{\theta}} s_t\|_2 + \|\nabla_a \hat{f} \nabla_{\boldsymbol{\theta}} \hat{a}_t - \nabla_a f \nabla_{\boldsymbol{\theta}} a_t\|_2$$
$$\leq \epsilon_f \|\nabla_{\boldsymbol{\theta}} \hat{s}_t\|_2 + L_f \|\nabla_{\boldsymbol{\theta}} \hat{s}_t - \nabla_{\boldsymbol{\theta}} s_t\|_2 + L_f \|\nabla_{\boldsymbol{\theta}} \hat{a}_t - \nabla_{\boldsymbol{\theta}} a_t\| + \epsilon_f \|\nabla_{\boldsymbol{\theta}} \hat{a}_t\|_2$$
$$\leq \epsilon_f \|\nabla_{\boldsymbol{\theta}} \hat{s}_t\|_2 + (L_f + L_f L_\pi) \|\nabla_{\boldsymbol{\theta}} \hat{s}_t - \nabla_{\boldsymbol{\theta}} s_t\|_2 + \epsilon_f \|\nabla_{\boldsymbol{\theta}} \hat{a}_t$$
$$= \epsilon_f \|\nabla_{\boldsymbol{\theta}} \hat{x}_t\|_2 + (L_f + L_f L_\pi) \|\nabla_{\boldsymbol{\theta}} \hat{s}_t - \nabla_{\boldsymbol{\theta}} s_t\|_2$$

Hence, applying this recursion we obtain that

$$\|\nabla_{\boldsymbol{\theta}} \hat{x}_{t+1} - \nabla_{\boldsymbol{\theta}} x_{t+1}\|_2 \leq \epsilon_f \sum_{k=0}^t (L_f + L_f L_\pi)^{t-k} \|\nabla_{\boldsymbol{\theta}} \hat{x}_k\|_2 \leq \epsilon_f \frac{L^{t+1} - 1}{L - 1} K^t$$

where $L = L_f + L_f L_\pi$. Then, the error in the gradient in the previous term is bounded by

$$\|\nabla_{\boldsymbol{\theta}}(Q - \hat{Q})\|_2 \leq \epsilon_Q K^H + L_Q \epsilon_f \frac{L^H - 1}{L - 1} K^H$$

In order to bound the model term we need first to bound the rewards since

$$\|\nabla_{\boldsymbol{\theta}}(M - \hat{M})\|_2 \leq \sum_{t=0}^H \gamma^t \|\nabla_{\boldsymbol{\theta}}(r(s_t) - r(\hat{s}_t))\|_2$$

Similar to the previous bounds, we can bound now each reward term by

$$\|\nabla_{\boldsymbol{\theta}}(r(s_t) - r(\hat{s}_t))\|_2 \leq \epsilon_f L_r \frac{L^{t+1} - 1}{L - 1} K^t$$

With this result we can bound the total error in models

$$\|\nabla_{\boldsymbol{\theta}}(M - \hat{M})\|_2 \leq \sum_{t=0}^{H-1} \gamma^t \epsilon_f L_r \frac{L^{t+1} - 1}{L - 1} K^t = \frac{L \epsilon_f}{(L - 1)} \left( \frac{(\gamma KL)^H - 1}{\gamma KL - 1} - \frac{(\gamma K)^H - 1}{\gamma K - 1} \right)$$

Then, the gradient error has the form

$$\mathbb{E}[\|\nabla_{\boldsymbol{\theta}} J_\pi(\boldsymbol{\theta}) - \nabla_{\boldsymbol{\theta}} \hat{J}_\pi(\boldsymbol{\theta})\|_2] \le \frac{L\epsilon_f}{(L-1)} \left( \frac{(\gamma KL)^H - 1}{\gamma KL - 1} - \frac{(\gamma K)^H - 1}{\gamma K - 1} \right) + \epsilon_Q(\gamma K)^H + L_Q\epsilon_f \frac{L^H - 1}{L-1}(\gamma K)^H$$

$$= \epsilon_f c_1(H) + \epsilon_Q c_2(H)$$

## A.2 PROOF OF LEMMA 4.2

The total variation distance can be bounded by the KL-divergence using the Pinsker's inequality

$$D_{TV}(\pi_{\boldsymbol{\theta}} \| \pi_{\hat{\boldsymbol{\theta}}}) \le \sqrt{\frac{D_{KL}(\pi_{\boldsymbol{\theta}} \| \pi_{\hat{\boldsymbol{\theta}}})}{2}}$$

Then if we assume third order smoothness on our policy, by the Fisher information metric theorem then

$$D_{KL}(\pi_{\boldsymbol{\theta}} \| \pi_{\hat{\boldsymbol{\theta}}}) = \tilde{c}\|\boldsymbol{\theta} - \hat{\boldsymbol{\theta}}\|_2^2 + \mathcal{O}(\|\boldsymbol{\theta} - \hat{\boldsymbol{\theta}}\|_2^3)$$

Given that $\|\boldsymbol{\theta} - \hat{\boldsymbol{\theta}}\|_2 = \alpha\|\nabla_{\boldsymbol{\theta}} J_\pi - \nabla_{\boldsymbol{\theta}} \hat{J}_\pi\|_2$, for a small enough step the following inequality holds

$$D_{KL}(\pi_{\boldsymbol{\theta}} \| \pi_{\hat{\boldsymbol{\theta}}}) \le \alpha^2 \tilde{c}(\epsilon_f c_1(H) + \epsilon_Q c_2(H))^2 =$$

Combining this bound with the Pinsker inequality

$$D_{TV}(\pi_{\boldsymbol{\theta}} \| \pi_{\hat{\boldsymbol{\theta}}}) \le \alpha\sqrt{\frac{\tilde{c}}{2}}(\epsilon_f c_1(H) + \epsilon_Q c_2(H)) = \alpha c_3(\epsilon_f c_1(H) + \epsilon_Q c_2(H))$$

## A.3 PROOF OF THEOREM 4.1

Given the bound on the total variation distance, we can now make use of the monotonic improvement theorem to establish an improvement bound in terms of the gradient error. Let $J_\pi(\boldsymbol{\theta})$ and $J_\pi(\hat{\boldsymbol{\theta}})$ be the expected return of the policy $\pi_{\boldsymbol{\theta}}$ and $\pi_{\hat{\boldsymbol{\theta}}}$ under the true dynamics. Let $\rho$ and $\hat{\rho}$ be the discounted state marginal for the policy $\pi_{\boldsymbol{\theta}}$ and $\pi_{\hat{\boldsymbol{\theta}}}$, respectively

$$\begin{aligned} |J_\pi(\boldsymbol{\theta}) - J_\pi(\hat{\boldsymbol{\theta}})| &= |\sum_{s,a} \rho(s)\pi_{\boldsymbol{\theta}} r(s,a) - \hat{\rho}(s)\pi_{\hat{\boldsymbol{\theta}}} r(s,a)| \\ &\le |\sum_{s,a} \rho(s)\pi_{\boldsymbol{\theta}}(a|s) r(s,a) - \hat{\rho}(s)\pi_{\hat{\boldsymbol{\theta}}}(a|s) r(s,a)| \\ &\le r_{\max} |\sum_{s,a} \rho(s)\pi_{\boldsymbol{\theta}}(a|s) - \hat{\rho}(s)\pi_{\hat{\boldsymbol{\theta}}}(a|s)| \\ &\le \frac{2r_{\max}}{1-\gamma} \max_s \sum_a |\pi_{\boldsymbol{\theta}}(a|s) - \pi_{\hat{\boldsymbol{\theta}}}(a|s)| \\ &= \frac{2r_{\max}}{1-\gamma} \max_s D_{TV}(\pi_{\boldsymbol{\theta}} \| \pi_{\hat{\boldsymbol{\theta}}}) \end{aligned}$$

Then, combining the results from Lemma 4.2 we obtain the desired bound.

## A.4 ABLATIONS

In order to show the significance of each component of MAAC, we conducted more ablation studies. The results are shown in Figure 5. Here, we analyze the effect of training the $Q$-function with data coming from just the real environment, not learning a maximum entropy policy, and increasing the batch size instead of increasing the amount of samples to estimate the value function.

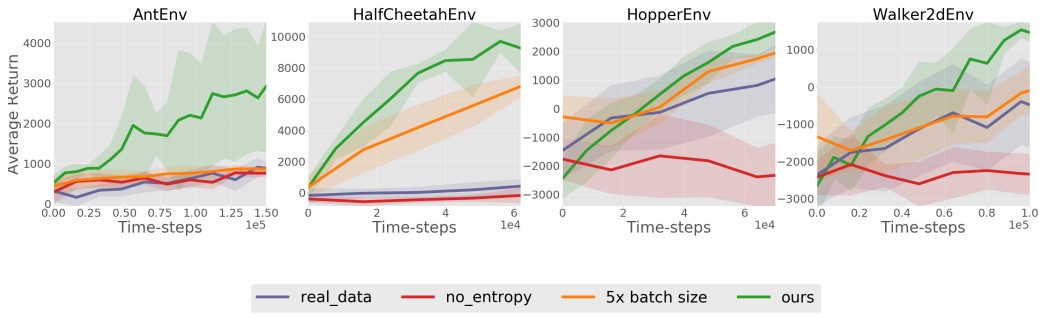

Figure 5: We further test the significance of some components of our method: not use the dynamics to generate data, and only use real data sampled from environments to train policy and Q-functions (real_data), remove entropy from optimization objects (no_entropy), and using a single sample estimate of the pathwise derivative but increase the batch size accordingly (5x batch size). Considering entropy and using dynamic models to augment data set are both very important.

## A.5 EXECUTION TIME COMPARISON

|  | Iteration (s) | Training Model (s) | Optimization (s) | MBPO Iteration (s) |
|---|---|---|---|---|
| HalfCheetahEnv | 1312 | 486 | 738 | 708 |
| HopperEnv | 845 | 209 | 517 | 723 |

Table 2: This table shows the time that different parts of MAAC need to train for one iteration after 6000 time steps, averaged across 4 seeds. We also add the time needed for MBPO for one iteration here for comparison.

