# OpenReview forum: "Model-Augmented Actor-Critic: Backpropagating through Paths"
_ICLR.cc/2020/Conference — Accept (Poster)_

### Official Review · AnonReviewer1 · 2019-10-13
**Official Blind Review #1**

**Rating:** 8

**Review:**

After seeing the clarifications made to address the other reviewers and my own reservations, I lean towards accepting this paper.  It is a simple yet novel way to incorporate a model, and I appreciate the thorough results with multiple baselines and additional ablations to show importance of each component of the method.

-------------------------------------

The authors suggest back-propagating through a learned dynamics model and provide a derivation on monotonic improvement of the proposed objective. They show increased sample efficiency, asymptotic performance matching model-free methods, and ability to scale to long horizons. They provide bounds on the error of the gradient when using the learned model and Q function and the total variation distance between policies trained using true dynamics versus the learned dynamics model. This leads to a theorem showing monotonic improvement of the policy under their algorithm, MAAC.

Decision: Weak Accept. The work contains both theoretical and empirical results on back-propagating through a learned dynamics model compared to other model-based and model-free methods. However, I'm unsure about the novelty of the method itself. How is this different from other planning through back propagation methods? This should be an additional section in related work. As examples, there is Universal Planning Networks [1], Differentiable MPC [2], and Path Integral Networks [3], [4] present a way to differentiate through path integral optimal control. I will increase my score if these concerns are addressed.

Nits:
Missing C in "Contrary to these methods" in Related Work section
Conclusion: analized -> analyzed

[1] Universal Planning Networks - Srinivas et al. 2018
[2] Differentiable MPC for End-to-end Planning and Control - Amos et al. 2018
[3] Path integral networks: End-to-end differentiable optimal control - Okada et al. 2017
[4] Mpc-inspired neural network policies for sequential decision making - Pereira et al. 2018

**Experience Assessment:**

I have read many papers in this area.

**Review Assessment: Checking Correctness Of Derivations And Theory:**

I assessed the sensibility of the derivations and theory.

**Review Assessment: Checking Correctness Of Experiments:**

I assessed the sensibility of the experiments.

**Review Assessment: Thoroughness In Paper Reading:**

I read the paper at least twice and used my best judgement in assessing the paper.

---

> ### Author Response · Authors · 2019-11-15
> **Response to Reviewer 1**
>
> We thank the reviewer for their thoughtful and constructive comments. As suggested by the reviewers, we ran additional experiments. The additional experiments and modifications are: 1) Added the SVG(1) as a baseline, 2) Extend the ablation study: MAAC w/o the entropy, increasing batch-size, and train with just real-data; 3) Extend the related work section, and 4) add more details and clarifications suggested by the reviewers, such as the learning of the Q-function.
> In the following, we address the specific comments of the reviewer:
>
> We have added an extra section on the related work suggested by the reviewer. The works suggested by the reviewer learn a differientable MPC planner. Instead, our method learns a neural network policy in an actor-critic fashion, using the model to extract more learning signal.
> Recent work [5], has demonstrated that these approaches achieve better performance and scale better than the MPC ones like [6]. For instance, [2, 3, 4] are evaluated in easier, low-dimensional domains. In our study, we evaluate the benefit of carrying out MPC on top of our learned policy at test time (section 5.4). The results suggest that the policy captures the optimal sequence of action, and re-planning does not result in significant benefits.
>
> We hope that with these clarifications and further analysis of our method the reviewer will consider our work for acceptance.
>
> [5] When to Trust Your Model: Model-Based Policy Optimization. Michael Janner, Justin Fu, Marvin Zhang, Sergey Levine.
> [6] Deep Reinforcement Learning in a Handful of Trials using Probabilistic Dynamics Models. Kurtland Chua, Roberto Calandra, Rowan McAllister, Sergey Levine

---

### Official Review · AnonReviewer2 · 2019-10-23
**Official Blind Review #2**

**Rating:** 6

**Review:**

This paper presents MAAC, a model-based reinforcement learning algorithm which makes use of path-wise derivative to optimize the policy. The learned Q function is used to estimate the terminal rewards and reduce instability and is trained by a STEVE-like style and Clipped Double Q. The dynamics model is learned by the same method in PETS. The policy is trained by directly computing the gradient to improve model-boosted Q. The theory shows that if learned model and learned Q function have similar derivative as the real one, the improvement of policy can be lower bounded. Experiments show that MAAC achieves the state-of-the-art and can be further improved by adding MPC.


The paper is well-written in general. The emperical results are very good and are claimed to be the new state-of-the-art.


Questions:

1. Could you please elaborate how J_Q(phi) is calculated? I found one in preliminary but I suppose the one used in Algorithm 1 is different as the paper states that MAAC also uses the model to train Q.
2. How is H(pi_theta) estimated? What's underlying state distribution of H(pi_theta)?
3. The objective of policy network is similar to SAC's objective, but the optimization is different. SAC optimizes the objective by minimizing the KL divergence between pi and exp(Q). Do you have any comparison between BPTT and KL divergence minimization?
4. How do you "use the cross-entropy method with our stochastic policy as our initial distributions" in MPC?
5. Does MAAC work for Humanoid?
6. How many samples are used to estimate the expectation in J_pi(theta)? For the single sample experiment, what will happen if the batch size is increased?
7. In Algorithm 1, how does "Sample trajectories T from \hat{f}_\phi" work?
8. To my understanding, when H = 0, the policy optimization doesn't use the model. Assuming Q function optimization uses the same H, the only usage of learned dynamics model is that pi and Q are trained in a different state distribution (D vs D_env). The ablation study shows that in this case, MAAC still outperforms SAC. So is it possible that D_model provides data augmentation?

**Experience Assessment:**

I have published one or two papers in this area.

**Review Assessment: Checking Correctness Of Derivations And Theory:**

I assessed the sensibility of the derivations and theory.

**Review Assessment: Checking Correctness Of Experiments:**

I assessed the sensibility of the experiments.

**Review Assessment: Thoroughness In Paper Reading:**

I read the paper at least twice and used my best judgement in assessing the paper.

---

> ### Author Response · Authors · 2019-11-15
> **Response to Reviewer 2**
>
> 1.We have extended the method section to detail how we learn J_Q(phi). It is learned by minimizing the bellman error, using all the real data seen so far and imagined one-step transitions of the model.
> 2.H(pi_theta) is estimated by taking the expectation among states and actions of -log(\pi(a|s)). Where \pi is the distribution resulting of applying tanh to a Gaussian r.v
> 3.We have this comparison on the ablation section, H=0 can be interpreted as KL minimization. It corresponds on using the exact same optimization as MAAC, but doing the policy update as in SAC.
> 4.Since MAAC learns a policy, a transition dynamics, and a Q-function, you can do CEM by sampling initially from the policy. That means, that at each time-step in the first iteration of CEM you sample from \pi(a|s) instead of a unit Gaussian. Then you evaluate the sequences of action using your learned model and the Q-function as terminal value.
> 5.Unfortunately, because of time constraints, we have not been able to evaluate humanoid until convergence. However, similar methods to ours do learn successfully on humanoid [1]. We expect MAAC to also achieve maximum performance on the humanoid task. We will add these results on the final.
> 6.We use a batch size of 256 and 5 samples to estimate the value function. We have added in the appendix the experiment requested of increasing the batch size x5. The results suggest that having a more accurate estimate of the value function results in better results than simply increasing the batch size.
> 7.To sample trajectory from \hat{f}_\phi, we sample a state s from our replay buffer and an action a from our policy ( a ~ \pi(a|s)). Then, we use our model to predict the next state s’. We repeat the procedure with s’ instead of s, until we get a sequence of states and actions. These are then stored into a replay buffer.
> 8.Yes, that is correct. D_model provides data augmentation, and this has been shown to decrease the sample complexity of SAC [1].

---

### Official Review · AnonReviewer3 · 2019-10-26
**Official Blind Review #3**

**Rating:** 3

**Review:**

#rebuttal responses
Thanks for the clarification!  However, I will keep my original score for these reasons:
(1) Only 3 random seeds are used for each environment, which is not convincing as the variance of MAAC is large in some figures.
(2) Baselines are only trained with 10^5 steps and do not converge. Thus it not fair to say that MAAC matches the asymptotic performance of model-free algorithms.

#review
This paper constructs a model-based policy optimization algorithm (MAAC) that uses the pathwise derivative of the learned model and policy across future timesteps. The terminal value function is used to improve stability.  The theoretical guarantee of the error of the model-based gradient is presented. Experimental results show that MAAC outperforms SAC, STEVE, and MBPO on four environments in terms of the sample efficiency.

The experimental results are strong and I appreciate the plots of the gradient error in a simple task, shown in Figure 3. But I want to see the comparison of the final performance of each algorithm in these environments, and I doubt that the baselines do not converge.

However, the paper is badly written. First of all, the authors claim that the pathwise derivate method is applied to optimize the objective function. But the detail of the method is missing. Secondly, I can not follow the procedure of how Q function is learned in the MAAC algorithm. Thirdly, Theorem 4.1 gives a performance improvement bound of the new policy w.r.t J_pi without the entropy term. But the MAAC algorithm optimizes the objective with the policy entropy term.

Also, MAAC applies many techniques in other papers. The paper does not clearly show the advantage of each component:
(1) I want to see the experimental results of MAAC optimizing the objective without the entropy.
(2) The SVG(1) algorithm should be compared as a baseline as the SVG(1) also uses the gradient of the learned model to optimize the policy.
(3) The policy and the Q function are optimized on both the real samples and the generated samples. I want to see the ablation study or justification on whether training on real samples or both samples.

Questions:
1. How many independent runs are used in experiments?
2. Does the computation of the pathwise derivate method cost much time？ Is MAAC much slower than SAC?

I am happy to increase my score if the authors justify these questions.

**Experience Assessment:**

I have published one or two papers in this area.

**Review Assessment: Checking Correctness Of Derivations And Theory:**

I assessed the sensibility of the derivations and theory.

**Review Assessment: Checking Correctness Of Experiments:**

I carefully checked the experiments.

**Review Assessment: Thoroughness In Paper Reading:**

I read the paper thoroughly.

---

> ### Author Response · Authors · 2019-11-15
> **Response to Reviewer 3**
>
> We thank the reviewer for their thoughtful and constructive comments. As suggested by the reviewers, we ran additional experiments. The additional experiments and modifications are: 1) Added the SVG(1) as a baseline, 2) Extend the ablation study: MAAC w/o the entropy, increasing batch-size, and train with just real-data; 3) Extend the related work section, and 4) add more details and clarifications suggested by the reviewers, such as the learning of the Q-function.
> In the following, we address the specific comments of the reviewer:
>
> Comments:
> >Detail of the algorithm is missing.
> We have extended the method section to include the details of our final algorithm.
>
> >How the q-function is learned.
> The Q-function is learned by minimizing the Bellman error with data from the real environment and samples from the model. To compute the targets we follow the procedure described in [1], which uses bootstrapped values to minimize the error of the targets.The effect of using [1] in our method is ablated in the experiment section. We have clarified this in the paper.
>
> >Theorem 4.1 performance bound of without the entropy term
> Our theorem gives a bound w.r.t the error on the gradient error, where the error comes from approximating the Q-function and the model. The entropy just depends on the policy, not on the Q function or the model. As a result, the gradient of the entropy on the J_\pi and \hat{J}_\pi is the same and cancels out when computing the error. Thus, our error bound also holds when learning maximum entropy policies.
>
> Experiments:
> 1. We have added to the appendix an extended ablation analysis that contains MAAC without the entropy. The results show that the entropy term is crucial for proper learning, as the original SAC [1] paper also demonstrates.
> 2. We implemented and ran SVG(1) as baseline, obtaining inferior performance to the other methods.
> 3. In the extended ablation of the appendix, we also show the performance of MAAC when the Q function and policy are trained on real data. The results show that augmenting the real-world data with imagined transitions from the model is crucial for sample efficient learning.
>
> Questions:
> 1. We used 3 random seeds for each experiment (our method, baselines, and ablations).
> 2. MAAC is slower than SAC. However, the main extra time does not come from backpropagating through the model. It is due to training the model and from being able to take more gradient steps per environment interaction thanks to the model. We added on the Appendix.
>
> We hope that with these clarifications and further analysis of our method the reviewer will consider our work for acceptance.
>
> [1] Soft Actor-Critic: Off-Policy Maximum Entropy Deep Reinforcement Learning with a Stochastic Actor. Tuomas Haarnoja, Aurick Zhou, Pieter Abbeel, Sergey Levine.

---

### Decision · Program_Chairs · 2019-12-19

**Decision:**

Accept (Poster)

**Comment:**

The authors propose a novel model-based reinforcement learning algorithm. The key difference with previous approaches is that the authors use gradients through the learned model. They present theoretical results on error bounds for their approach and a monotonic improvement theorem. In the small sample regime, they show improved performance over previous approaches.

After the revisions, reviewers raised a few concerns:
The results are only for 100,000 steps, which does not support the claim that the models achieves the same asymptotic performance as model – free algorithms would.
The results would be stronger as the experiments were run with more than 3 random seats.
In the revised version of the text, it's unclear if the authors are using target networks.

Overall, I think the paper introduces some interesting ideas and shows improved performance over existing approaches. I recommend acceptance on the condition that the authors tone down their claims or back them up with empirical evidence. Currently, I don't see evidence for the claim that the method achieves similar asymptotic performance to model free algorithms or the claim that their approach allows for longer horizons than previous approaches.